# Synthesis, X-ray Analysis, Biological Evaluation and Molecular Docking Study of New Thiazoline Derivatives

**DOI:** 10.3390/molecules24091654

**Published:** 2019-04-26

**Authors:** Yahia N. Mabkhot, H. Algarni, Abdulrhman Alsayari, Abdullatif Bin Muhsinah, Nabila A. Kheder, Zainab M. Almarhoon, Faiz A. Al-aizari

**Affiliations:** 1Department of Pharmaceutical Chemistry, College of Pharmacy, King Khalid University, Abha 61441, Saudi Arabia; 2Department of Physics, Faculty of Sciences, King Khalid University, P.O. Box 9004, Abha 61441, Saudi Arabia; halgarni@kku.edu.sa; 3Research Centre for Advanced Materials Science (RCAMS), King Khalid University, P.O. Box 9004, Abha 61441, Saudi Arabia; 4Department of Pharmacognosy, College of Pharmacy, King Khalid University, Abha 61441, Saudi Arabia; alsayari@kku.edu.sa (A.A.); ajmohsnah@kku.edu.sa (A.B.M.); 5Department of Chemistry, Faculty of Science, Cairo University, Giza 12613, Egypt; nabila.abdelshafy@gmail.com; 6Department of Chemistry, College of Science, King Saud University, P.O. Box 2455, Riyadh 11451, Saudi Arabia; zalmarhoon@ksu.edu.sa (Z.M.A.); faizalaizari@yahoo.com (F.A.A.-a.); 7Department of Chemistry, Faculty of Science, AL-Baydha University, Albaydah 38018, Yemen

**Keywords:** thiazoline, X-ray crystallography, molecular docking, antimicrobial activity, cytotoxic activity

## Abstract

A series of new thiazoline derivatives were synthesized. Structure analyses were accomplished employing ^1^H-NMR, ^13^C-NMR, X-ray and MS techniques. The in vitro antitumor activities were assessed against human hepatocellular carcinoma (HepG-2) and colorectal carcinoma (HCT-116) cell lines. The results revealed that the thiazolines **5b** and **2c** exhibited significant activity against the two cell lines. The in vitro antimicrobial screening showed that the thiazolines **2c**, **5b** and **5d** showed promising inhibition activity against *Salmonella* sp. Additionally, the inhibition activity of thiazolines **2e** and **5b** against *Escherichia coli* was comparable to that of the reference compound gentamycin.

## 1. Introduction

Thiazoline-based compounds possess a variety of biological activities such as antiproliferative [1,2,3,4,5,6,7,8,9,10], anti-inflammatory [11], antimicrobial, [12] and antioxidant properties [13]. In addition, they are reported as butyrylcholinesterase and carboxylesterase inhibitors [14]. Motivated by the above-mentioned results, numerous design and synthesis efforts have been employed to develop new derivatives with more effective and safer therapeutic profiles [15,16,17,18,19,20,21,22,23,24,25,26,27,28,29,30]. Additionally, heterocycles based on a thiazoline-2-thione core may undergo several chemical reactions, including alkylation, oxidation, and cycloaddition, as a result of having two different nitrogenous and sulfurous groups [31,32,33,34,35,36,37]. There are many methods for preparing thiazoline-2-thione derivatives from alkyl ammonium dithiocarbamates with the appropriate α-halocompound in the presence of an acid [38,39,40,41,42] or by using primary amines, CS_2_ and α-haloketone in DMF [43] or toluene [44] or in water as a solvent [45], or without solvent or catalyst [46]. These findings encouraged us to conduct a slightly modification of the reported reaction condition [45,47,48] to synthesize new thiazoline derivatives and to investigate their synthetic potential in the preparation of new thiazoline based heterocycles in order to assess their biological activities. 

## 2. Results and Discussion

### 2.1. Chemistry

The first series consisted of stirring 3-chloropentane-2,4-dione (**1a**) or ethyl 2-chloro-3-oxobutanoate (**1b**) with the appropriate primary amine and carbon disulfide in ethanol at room temperature; this afforded the target thiazoline-2-thione derivatives **2a**–**e** in high yields (Scheme 1).

The suggested mechanism for their synthesis is illustrated in Scheme 2. In this mechanism, the treatment of aniline with carbon disulfide results ammonium dithiocarbamate **3**, which reacts with α-halo-1,3-diketone **1** to produce acyclic dithiocarbamate derivative **4**. Intramolecular cyclization of intermediate **4**, followed by dehydration, yields the target compounds **2a**–**e** (Scheme 2).

The structures of the obtained products **2a**–**e** were established and confirmed via spectroscopic (NMR, MS, IR) and elemental analyses. Their ^1^H-NMR spectra showed the presence of a singlet signal due to methyl protons at carbon 4, in addition to the other expected proton signals. In addition, their ^13^C-NMR spectra confirmed the assigned structures and revealed the presence of the expected signals of carbonyl and thiocarbonyl at δ 177.5–188.17 and δ 190.0–193.18, respectively (see the Materials and Methods section). It is important to note that compounds **2a**–**d** were prepared by another two methods as follows: (i) in 54% and 74% yield, respectively, by using the same reagents as for compounds **2a**,**b** in the presence of NaOH as basic catalyst [47]; (ii) in yields of 74% and 75%, respectively, by the second literature method for synthesis of derivatives **2c**,**d** from α-halo-compounds with ammonium dithiocarbamate [48]. Our method for synthesis of compounds **2a**–**e** features better yields of these compounds. 

Refluxing compound **2a** with the appropriate aniline derivatives afforded a new series of thiazoline derivatives, **5a**–**e**, as shown in Scheme 3. The structures of the target products **5a**–**e** were deduced from their IR, NMR and mass spectra. For example, their IR spectra revealed the absence of any absorption band due to the carbonyl group, which was apparent in compound **2a**. Also, the ^13^C-NMR spectra of the synthesized products **5a**–**e** showed, in each case, the absence of the carbonyl signal (see the Materials and Methods section). Moreover, the structure of **5c**, as a typical example of the prepared series, was confirmed by single crystal X-ray analysis (Figure 1). 

Next, the reaction of the thiazoline-2-thione derivative **2a** [45,47] with 2-oxo-*N*′-phenylpropane hydrazonoyl chloride (**6**) [49,50] produced the spiro-compound **7 [51]** (Scheme 3). The reaction was assumed to proceed via a 1,3-dipolar cycloaddtion reaction between nitrile imine (formed in situ from hydrazonoyl halide **6** by the action of triethylamine) and C=S of thiazoline-2-thione derivative **2a**. The ^1^H-NMR spectrum of spiro-compound **7** presented three signals at δ 1.92, 2.10 and 2.22, corresponding to three sets of methyl group protons. Additionally, its spectrum showed signals of the phenyl ring in the 7.35–7.59 ppm region. Also, the signal of C=S disappeared in its ^13^C-NMR spectrum.

### 2.2. X-ray Analysis

The crystallographic data of thiazoline derivative **5c** and the refinement information are summarized in Table 1. The selected bond lengths and bond angles are listed in Table 2. The asymmetric unit contains one independent molecule, as shown in Figure 1. All the bond lengths and angles are in normal ranges [52]. In the crystal packing, shown in Figure 2, molecules are linked via one intermolecular hydrogen bond (Table 3).

The Crystallographic data for thiazoline **5c** (CCDC Number 1818811) can be obtained on request from the director, Cambridge Crystallographic Data Center, 12 Union Road, Cambridge CB2 1EW, UK.

### 2.3. Biological Evaluation

#### 2.3.1. The In Vitro Antimicrobial Assessment of the Synthesized Thiazolines

The antifungal and antibacterial potency of the synthesized compounds and the reference drugs were evaluated against two fungal species (*Aspergillus fumigatus* (RCMB 002008 (4) and *Candida albicans* (RCMB 05036)), two gram positive bacteria *(Staphylococcus aureus* (RCMB010010) and *Bacillus subtilis* (RCMB 010067)), and two gram negative bacteria (*Salmonella* sp. (RCMB 010043) and *Escherichia coli* (RCMB 010052)) using the inhibition zone technique according to the reported methods [53,54]. The results of this assessment are depicted in Table 4. Tests indicate that the compounds **2c**, **2d**, **2e**, and **5d** had significant antifungal activity against *Aspergillus fumigatus*. Additionally, all the evaluated thiazolines, except **2b**, were effective against *Candida albicans*. The study also showed that the tested compounds had important biological effectiveness against *Staphylococcus aureus* and *Bacillus subtilis* (except **2b**). The evaluation results showed that all the test compounds were effective against *Salmonella* sp., particularly **2c**, **5b** and **5d**; they approach the potency of Gentamycin. Moreover, the inhibition potency of thiazolines **2e** and **5b** is similar to the potency of Gentamycin towards *Escherichia coli*.

#### 2.3.2. Molecular Docking

The docking study play important role in predicting with the biological activity of any compounds. So, this study is considered a key for design and manufacturing of new drugs. We selected two derivatives **2c** and **5b** from the two synthesized series to use in molecular docking to study their behavior and their mode of action. Both thiazoline derivatives **2c** and **5b** were docked with human cyclin-dependent kinase enzyme (CDK 2), one of the kinase family, due to their important role in cell meiosis and replication.

Molecular docking was implemented using the MOE 2014.010 Package software. The structure of CDK2 was obtained from a protein data bank. Protein was optimized by adding hydrogen, repairing broken amino acid residues and removing water. In addition, compounds **2c** and **5b** were optimized for docking by adding hydrogen and then forcing energy minimization. 

The binding affinity of thiazoline **2c** showed hydrogen acceptor interactions with **Thr 14** and **Lys 129**, with binding energies equal −4 (kcal/mol). Also, it exhibited a pi-hydrogen interaction with **Gly 13**, with binding energy equal −0.6 (kcal/mol) (Figure 3).

In contrast, thiazoline derivative **5b** had less binding affinity to the CDK 2 enzyme than compound **2c**. It showed only one pi-hydrogen interaction with **Glu 12**, with binding energy equal -0.1 (kcal/mol) (Figure 4).

#### 2.3.3. Antitumor Evaluation of Some Selected Examples of the Synthesized Compounds

The in vitro antitumor activity of some selected examples of the synthesized thiazolines and the reference drug Doxorubicin were investigated using the MTT method [55] against human hepatocellular carcinoma cell line (HepG-2) and colon carcinoma cells (HCT-116). The concentration of the tested thiazolines needed to inhibit 50% of the cells population (IC_50_) was calculated and is presented in Table 5 and Table 6 and Figure 5 and Figure 6. 

The results of Table 5 and Table 6 showed that the tested thiazolines **5b** and **2c** have the highest effectiveness compared to the other thiazoline derivatives against the HepG-2 and HCT-116 cell lines, with IC_50_ values of approximately 7.46 µg/mL and 27.9 µg/mL for HepG-2 and 12.6 µg/mL and 29.1 µg/mL, for HCT-116, respectively. The remaining compounds have a noticeably moderated efficiency.

## 3. Materials and Methods 

### 3.1. Chemistry 

#### 3.1.1. General Information

All the melting points were measured using a Gallenkamp apparatus (Thermo Fisher Scientific, Paisley, UK) in open glass capillaries and are uncorrected. Infrared spectra (IR) were recorded using the KBr disc technique on a Perkin Elmer FT-IR spectrophotometer 1000 (PerkinElmer, Waltham, MA, USA). NMR spectra (^1^H and ^13^C) were measured using an ECP 400 NMR spectrometer (JEOL, Tokyo, Japan) operating at 400 MHz in deuterated chloroform (CDCl_3_). Mass spectra were measured on a Shimadzu GCMS-QP 1000 EX mass spectrometer (Tokyo, Japan) at 70 eV. Elemental analysis were recorded on a 2400 CHN Elemental Analyzer. The single-crystal X-ray diffraction measurements were done on a SMART APEX II CCD diffractometer (Bruker AXS Advanced X-ray Solutions GmbH, Karlsruhe, Germany). The biological assessments of the synthesized compounds were done in the Medical Mycology Laboratory of the Regional Center for Mycology and Biotechnology of Al-Azhar University, Cairo, Egypt. The 2-oxo-*N*′-phenylpropanehydrazonoyl chloride (**6**) was prepared as described in the literature [49,50].

#### 3.1.2. The Synthetic Procedure for the Target Thiazolines **2a**–**e**

To a solution of 3-chloropentane-2,4-dione (0.134 g, 0.112 mL, 1 mmol) or ethyl 2-chloro-3-oxobutanoate (0.164 g, 0.138 mL, 1 mmol) in ethanol (15 mL), carbon disulfide (0.152 g, 0.12 mL, 2 mmol) and the appropriate amine derivative (1 mmol) were added. The reaction mixture was stirred for six hours. The solid product that formed was filtered and washed with ethanol, and recrystallized to afford the corresponding thiazolines **2a**–**e**. The physical properties and spectroscopic data of compounds **2a**–**d** are in agreement with the literature [45,47,48].

*1-(4-Methyl-3-phenyl-2-thioxo-2,3-dihydrothiazol-5-yl)ethan-1-one* (**2a**): Yellow solid, (0.23 g, yield 92%); m.p. 172–174 °C (EtOH) [lit mp. 171–174 °C [45,47]]; IR v_max_ 1630 (C=O), 1490 (C=S) cm^−1^; ^1^H-NMR (CDCl_3_) δ 2.25 (s, 3H, CH_3_), 2.35 (s, 3H, CH_3_), 7.14–7.52 (m, 5H, Ph); ^13^C-NMR (CDCl_3_) δ 16.14, 30.23 (2CH_3_), 112.50, 147.34, 121.50, 128.05, 130.16, 137.08, 188.17, 190.00. Anal. Calcd. for C_12_H_11_NOS_2_ (249.35): C, 57.80; H, 4.45; N, 5.62. Found: C, 57.91; H, 4.52; N, 5.55%.

*Ethyl 4-methyl-3-phenyl-2-thioxo-2,3-dihydrothiazole-5-carboxylate* (**2b**): white powder, (0.24 g, yield 85%); m.p. 160–162 °C (EtOH) [lit. mp. 158–160 °C [47]]; IR v_max_ 1698 (C=O), 1621 (C=C), 1514 (C=S) cm^−1^; ^1^H-NMR (CDCl_3_) δ 1.29 (t, 3H, CH_3_), 2.25 (s, 3H, CH_3_), 4.25 (q, 2H, CH_2_), 7.15–7.53 (m, 5H, Ph); MS (*m/z*) (%) 279 (M^+^, 100%), 278 (54%), 250 (59%), 234 (12%). Anal. Calcd. for C_13_H_13_NO_2_S_2_ (279.38): C, 55.89; H, 4.69; N, 5.01. Found: C, 55.82; H, 4.73; N, 5.10%.

*1-(4-Methyl-2-thioxo-2,3-dihydrothiazol-5-yl)ethanone* (**2c**): white powder, (0.147 g, yield 85%); m.p. 208–210 °C (EtOH) [lit. mp. 210–211 °C [48]]; IR v_max_ 3210 (NH), 1669 (C=O), 1617(C=C), 1537(C=S) cm^−1^; ^1^H-NMR (CDCl_3_) δ 1.77 (s, 3H, CH_3_), 2.35 (s, 3H, CH_3_), 7.2 (s, H, NH); ^13^C-NMR (CDCl_3_) δ 17.70 (CH_3_), 25.60 (CH_3_), 119.00, 148.00, 188.00, 192.00; MS (*m/z*) (%) 173 (M^+^, 2%), 43 (34%). Anal. Calcd. for C_6_H_7_NOS_2_ (173.26): C, 41.59; H, 4.07; N, 8.08. Found: C, 41.63; H, 4.14; N, 8.12%.

*Ethyl 4-methyl-2-thioxo-2,3-dihydrothiazole-5-carboxylate* (**2d**): white powder, (0.152 g, yield 75%); m.p. 146–148 °C (EtOH) [lit. mp. 151–152 °C [48]]; IR v_max_ 3383 (NH), 1674 (C=O), 1601 (C=C), 1425 (C=S) cm^−1^; ^1^H-NMR (CDCl_3_) δ 1.36 (t, 3H, CH_3_), 2.49 (s, 3H, CH_3_), 4.35 (q, 2H, CH_2_), 7.19 (s, H, NH); ^13^C-NMR (CDCl_3_) δ 14.22 (CH_3_), 29.34 (CH_3_), 61.94 (CH_2_), 118.50, 162.81, 177.50, 193.18; MS (*m/z*) (%) 204 (16%), 203 (M^+^, 70%), 159 (99%), 158 (30%), 43 (100%). Anal. Calcd. for C_7_H_9_NO_2_S_2_ (203.28): C, 41.36; H, 4.46; N, 6.89. Found: C, 41.42; H, 4.53; N, 6.78%.

*Ethyl 4-methyl-3-(phenylamino)-2-thioxo-2,3-dihydrothiazole-5-carboxylate* (**2e**): white powder, (0.25 g, yield 85%); mp 210–212 °C (EtOH); IR v_max_ 3390 (NH), 1701 (C=O), 1593 (C=C), 1544 (C=S) cm^−1^; ^1^H-NMR (CDCl_3_) δ 1.33 (t, 3H, CH_3_), 2.45 (s, 3H, CH_3_), 4.30 (q, 2H, CH_2_), 7.21–7.69 (m, 5H, ArH), 11.85 (s, 1H, NH); MS (*m/z*) (%) 294 (M^+^, 1%), 248 (6%), 216 (80%), 77 (100). Anal. Calcd. for C_13_H_14_N_2_O_2_S_2_ (294.39): C, 53.04; H, 4.79; N, 9.52. Found: C, 53.12; H, 4.84; N, 9.43%. 

#### 3.1.3. Synthetic Procedure for Substituted 4-Methyl-3-Phenylthiazole-2(3*H*)-Thione **5a**–**e**

A mixture of the appropriate amine (1 mmol) and thiazoline **2a** (0.249 g, 1 mmol) in ethanol (10 mL) was refluxed for approximately six hours until the precipitation was produced. Then, the product was filtered and recrystallized from ethanol to afford the corresponding condensation product. 

*5-(1-Hydrazonoethyl)-4-methyl-3-phenylthiazole-2(3H)-thione* (**5a**): Yellow powder, (0.145 g, yield 55%); m.p. 215–217 °C (EtOH); IR v_max_ 3208, 3160 (NH_2_), 1631 (C=N), 1487 (C=S) cm^−1^; ^1^H-NMR (CDCl_3_) δ 1.99 (s, 3H, CH_3_), 2.49 (s, 3H, CH_3_), 6.65 (s, 2H, NH_2_), 7.30–7.60 (m, 5H, Ph); MS (*m/z*) (%) 264 (22%), 263 (M^+^, 82%),77(100%). Anal. Calcd. for C_12_H_13_N_3_S_2_ (263.38): C, 54.72; H, 4.97; N, 15.95. Found: C, 54.68; H, 4.87; N, 15.88%. 

*4-Methyl-3-phenyl-5-(1-(phenylimino)ethyl)thiazole-2(3H)-thione* (**5b**): Yellow powder, (0.162 g, yield 50%; m.p. 228–230 °C (EtOH); IR v_max_ 1590 (C=N), 1492 (C=S) cm^−1^; ^1^H-NMR (CDCl_3_) δ 1.95 (s, 3H, CH_3_), 2.17 (s, 3H, CH_3_), 7.30-7.59 (m, 10H, Ph); MS (*m/z*) (%) 324 (M^+^, 2%), 247 (24%), 77(100%). Anal. Calcd. for C_18_H_16_N_2_S_2_ (324.46): C, 66.63; H, 4.97; N, 8.63. Found: C, 66.55; H, 4.83; N, 8.55%. 

*(E)-4-Methyl-3-phenyl-5-(1-(2-phenylhydrazono)ethyl)thiazole-2(3H)-thione* (**5c**): Yellow powder, (0.203 g, yield 60%); m.p. 220–222 °C (EtOH); IR v_max_ 3289 (NH), 1589 (C=N), 1495 (C=S) cm^−1^; ^1^H-NMR (CDCl_3_) δ 2.08 (s, 3H, CH_3_), 3.32 (s, 3H, CH_3_), 7.12–7.58 (m, 10H, ArH), 9.51 (s, 1H, NH); ^13^C-NMR (CDCl_3_) δ 15.38 (CH_3_), 15.71 (CH_3_), 112.71, 137.96, 119.31, 124.02, 123.81, 128.57, 128.96, 134.30, 135.47, 137.96, 145.27, 186.91; MS (*m/z*) (%) 340 (20%), 339 (M^+^, 96%), 247 (14%), 77(100%). Anal. Calcd. for C_18_H_17_N_3_S_2_ (339.48): C, 63.68; H, 5.05; N, 12.38. Found: C, 63.73; H, 5.12; N, 12.29%. 

*5-(1-((1H-1,2,4-Triazol-5-yl)imino)ethyl)-4-methyl-3-phenylthiazole-2(3H)-thione* (**5d**): Yellow powder, (0.158 g, yield 50%); m.p. 275–277 °C (DMF); IR v_max_ 3370 (NH), 1637 (C=N), 1577(C=C), 1470 (C=S) cm^−1^; ^1^H-NMR (CDCl_3_) δ 1.95 (s, 3H, CH_3_), 2.20 (s, 3H, CH_3_), 7.30-7.62 (m, 6H, Ph), 8.72 (s, 1H, NH); MS (*m/z*) (%) 315 (M^+^, 6%), 300 (10%), 105 (68%). Anal. Calcd. for C_14_H_13_N_5_S_2_ (315.42): C, 53.31; H, 4.15; N, 22.20. Found: C, 53.28; H, 4.22; N, 22.12%. 

*5-(1-((1H-Benzo[d]imidazol-2-yl)imino)ethyl)-4-methyl-3-phenylthiazole-2(3H)-thione* (**5e**): Yellow powder, (0.164 g, yield 45%); m.p. 200–202 °C (EtOH); IR v_max_ 3236 (NH), 1586 (C=C), 1487 (C=S) cm^−1^; ^1^H-NMR (CDCl_3_) δ 1.90 (s, 3H, CH_3_), 2.25 (s, 3H, CH_3_), 7.20–7.61 (m, 9H, Ph), 11.62 (s, 1H, NH); MS (*m/z*) (%) 364 (M^+^, 8%), 249 (52%), 248 (36%), 133 (98%), 43 (100%). Anal. Calcd. for C_19_H_16_N_4_S_2_ (364.49): C, 62.61; H, 4.42; N, 15.37. Found: C, 62.55; H, 4.38; N, 15.42%. 

#### 3.1.4. 3,7-Diacetyl-8-methyl-1,9-diphenyl-4,6-dithia-1,2,9-triazaspiro[4.4]nona-2,7-diene (**7**)

To a mixture of thiazole derivative **2a** (0.249g, 1 mmol) and 2-oxo-*N*′-phenylpropane hydrazonoyl chloride (**6**) (0.196 g, 1 mmol) in dry benzene (10 mL), triethylamine (0.2 g, 0.28 mL, 2 mmol) was added and the reaction mixture was heated under reflux for 6 h. The precipitated triethylamine hydrochloride was removed by filtration and the filterate was evaporated under a vacuum. The remaining residue was treated using ethanol and the solid product that formed was filtered off and recrystallized using ethanol to give the spiro-compound **7**. Yellowish powder, (0.204 g, yield 50%); m.p. 154–156 °C (DMF)[Lit mp. 155–157 °C (MeCN)] [51]]; IR v_max_ 1644 (C=O), 1549 (C=C) cm^−1^; ^1^H-NMR (CDCl_3_) δ 1.92 (s, 3H, CH_3_), 2.10 (s, 3H, CH_3_), 2.22 (s, 3H, CH_3_), 7.35–7.59 (m, 10H, Ph); MS (*m/z*) (%) 409 (M^+^, 2%), 43 (100%). Anal. Calcd. for C_21_H_19_N_3_O_2_S_2_ (409.52): C, 61.59; H, 4.68; N, 10.26. Found: C, 61.66; H, 4.73; N, 10.22%.

### 3.2. X-ray Analysis 

The thiazoline derivative **5c** was obtained as single crystals by slow evaporation from an ethanol solution of the pure compound at room temperature. The thiazoline structure was evaluated using SHELXT [56,57]. All the crystallographic data of the crystal structure **5c** are available and can be obtained free of charge from the Cambridge Crystallographic Data Centre via www.ccdc.cam.ac.uk/data_request/cif. For additional details, refer to the Appendix A.

### 3.3. Biological Evaluations

#### 3.3.1. The In Vitro Antimicrobial Investigation

The antimicrobial activities of the newly synthesized thiazolines were evaluated by the inhibition zone technique on *Aspergillus fumigatus*, *Candida albicans*, *Staphylococcus aureus*, *Bacillus subtilis*, *Salmonella* sp. and *Escherichia coli* [53,54]. Additional details are available in the Appendix A.

#### 3.3.2. In Vitro Cytotoxic Activity

The cytotoxic assessment of the target thiazole derivatives was carried out against two cancer cell lines (HepG2 and HCT-116) using the MTT assay after 24 h of incubation [55]. The experimental procedure is included in the Appendix A.

### 3.4. Molecular Modeling

The docking study was performed using the MOE 2014.09 software [58]. Regularization and optimization for the protein and ligand were performed. Each docked thiazole was assigned a score according to its fit in the ligand binding pocket (LBP) and its binding mode.

## 4. Conclusions

In this work, new thiazolines were prepared, characterized and evaluated for their biological activities. The results of the antimicrobial evaluation indicated that the thiazoline derivatives **2c**, **5b** and **5e** exhibited high inhibitory activity against *Salmonella* sp., while compounds **2e** and **5b** were comparable to Gentamycin against *Escherichia coli*. The data from the in vitro antitumor evaluation revealed that the thiazolines **5b** and **2c** were the most effective against the HepG-2 and HCT-116 cell lines. This remarkable efficacy has potential usage in numerous pharmaceutical applications. Molecular docking study supported anticancer results and showed binding affinities for thiazolines **5b** and **2c** towards cyclin-dependant kinase 2.

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
