# Peer review of "Synthesis, X-ray Analysis, Biological Evaluation and Molecular Docking Study of New Thiazoline Derivatives"

_molecules, 2019, doi:10.3390/molecules24091654_

Round 1

Reviewer 1 Report

The paper entitled "A Convenient Synthesis, X-Ray Analysis, Biological Evaluation and Molecular Docking Study of New Thiazole Derivatives" by Yahia N, Mabkhot presents the synthesis of some thiazole derivaties and for some of them their biological activity investigation, including a molecular docking study. The research done by the authors could be interesting, however it has some serious flaws:

for starter, the title, why not simply "Synthesis, X-Ray Analysis...etc." instead of "A convenient synthesis....." when you don't point out at all the inconveniences. I myself have synthesized some thiazoles during my PhD and never had any problem with the synthesis;

the paper must be checked by an English expert;

lines 32-24 were "inspired" by 16 references, while this phrase doesn't say much, therefore it must be developed;

"in vitro" should be "in vitro";

replace "inhibition activity" with "inhibitory activity";

replace "Salmonella SP" with "Salmonella sp.";

replace "via" with "via";

on page 3, there are many "Error! Reference source not found" messages;

concerning the antibacterial activity investigation: what happened to compounds 2a, 7 and 8? why were they excluded?; why did you choose amphotericin B, ampicillin and gentamycin as reference drugs?; I also suggest MIC values determination, since the zone inhibition diameter determination is so before 2000;

concerning the antitumor activity investigation: what happened to compounds 2a, 2b, 5c, 7 and 8? why were they excluded?; in table 6 even 5a is missing? why did you use doxorubicin as standard drug?;

lines 153-156: the most active compound is 5b, which is 25 fold less active than the reference drug, so basically it has a very weak antitumor effect; therefore, saying about the other compounds, that are even less active that 5b, that have a "moderate efficiency" is simply exaggerated;

line 159: why did you conclude that the thiazole ring is essential for the antitumor activity, since non-thiazole derivatives were not included in this experiment?; the sentence "the substituents can increase or decrease reactivity" is very speculative and I believe you meant "activity";

line 161: there is no "attachement" of two phenyl rings; for example naphtaline or phenantrene have 2 phenyl rings attached;

line 164: how did you get to this conclusion, when the only compound that has a CH3CO group on the thiazole moiety is no. 7, and was apparently not tested?

why only compounds 2c and 5b were included in the molecular docking study? and why did you choose cyclin-dependent kinase as target? since you have used doxorubicine as referenece antitumor agent, I would suggest redoing this part of your research on topoisomerase-II, and including doxorubicine in the study;

the binding energies towards the enzyme (-4 and - 0.6) are very low, we could basically assume that there is no binding...;

and last, references: more than half of them are veeeery old, especially no.35; I am sure there are newer techniques and more recent reviews on this subject.

Author Response

Dear Editor:

Thank you for your E-mail concerning manuscript number (molecules-479281).  Attached herewith, please find our revised manuscript having complied with technical remarks of the reviewers. We have complied with all requested corrections. Key changes are noted in red in the revised Manuscript. Further, the authors provide detailed responses to referee comments, on the pages that follow this cover letter. These comments are also recorded in red. On behalf of the coauthors, I would like to thank the reviewers for their feedback and constructive comments

Reviewer 2 Report

In this paper the synthesis of 5-acetylthiazoline-2-thiones 2 via the one pot multicomponent reaction of 2-chloroacetoacetates, CS2 and anilines in EtOH is presented. The products are formed in good yields and are isolated by filtration. The representative thiazoline-2-thione 2a is used for further reactions which include (a) condensation at the C=O with anilines, (b) a cycloaddition at C=S with a nitrile imine, and (c) bromination at the carbonyl α-position. The biological properties for some of the synthesized compounds 2 and 5 are reported and the authors attempt to describe SAR. Docking studies for the most active compounds were performed.

While the authors report that the synthesized compounds are new, this is true only for the imine analogues 5. Compounds 2, 7 and 8 are reported in the literature, but no indication about this is given throughout the manuscript. The relevant references for all these compounds should be included at the appropriate places.

Thus, the only “novelty” that I can see from this study is (a) the slightly modified reaction conditions for obtaining the compounds 2 which seems to benefit from the other methods to that it is chromatography free and (b) the preparation of compounds 5 (although the chemistry for their preparation is trivial) and the studies on their biological activities (although I do not understand how the authors selected specific analogues for their antitumor studies and didn’t study all 5 analogues). But I don’t find the number of compounds synthesized and the biological data obtained satisfactory for attempting a SAR discussion.

Furthermore, I have the following criticisms:

Introduction

(a)   The introduction needs some strengthening. The following papers reporting the synthesis of thiazoline-2-thiones should be included in addition to the others: (1) J. Heterocyclic Chem. 2017, 54, 3600; (2) Phosphorus, Sulfur, and Silicon, 2011, 186, 12; and (3) J. Heterocyclic Chem. 1967, 4, 605.

(b)   In line 38-39: “only a few protocols succeed in forming the desired derivatives with a pure state and a high yield.” It might worth to mention the best ones and the limitations for the ones that do not work very well.

(c)   Statement in lines 39-41 is misleading since only compounds 5 are known. This needs to be corrected accordingly. Maybe rephrase to something like “thiazoline-2-thiones 2 are prepared using a slightly modified reaction conditions to those reported in the literature[ref] and further synthetic manipulation of the obtained compounds gave the new compounds 5 for which their biological activities were studied”

Schemes

Scheme 1: where there is R1 must change to R1; include in the table yields for the products.

Scheme 2: where there is R1 must change to R1

Scheme 3: Yields for the compds 5a-e must be included in the “table” and the yields, reaction time and temperature must be included in the arrows for the transformation to compounds 7 and 8. Conditions from 2a to 8 would look better if they are rotated +90°. In some cases, the terminal methyl is written as CH3 and in some cases is omitted, please change it to one style to be consisted. This applies for Scheme 2 too.

In Table 3: (a) N3—H1N3···S2 needs to be corrected to N3—H1···S2.

(b) the distances D-H, H···A should not be included. There is no physical meaning since X-ray does not locate the hydrogen atoms but only the heavier atoms. The hydrogen atoms are added by the program but are not the actual experimental distances. Only the D-A distance should be reported.

(c) change (Å, °) to (Å or °). Same for table 2.

Results and Discussions

2.1. Chemistry

(a)   For the reaction to obtain compounds 2, how did the authors conclude to these reaction conditions? Any optimisation? Any other solvents used? How does their conditions compare with the other literature methods? Are there any benefits? Would worth at this point to state how these conditions are better (if they are) to the others already reported.

(b)   In lines 57 -61 the spectroscopic characterisation of the compounds is mentioned. Would beneficial, where the 13C-NMR is discussed, to include some characteristic peaks that support the structural assignment such as the presence of the C=O and C=S signals, if possible.

(c)   Lines 71-77:similar reaction is presented in the literature and the relevant literature should be cited: Russian Journal of Organic Chemistry, 2005, 41, 308−310; and Russian Journal of Organic Chemistry, 2007, 43, 1516−1525.

2.3 Biological activity

I don’t know what guided the authors to study the antitumor activity for only the specific analogues. The study would significantly benefit if the antitumor activity for all five compounds 5a-e was studied. The compound 2a should be also included in the studies as it is the only one that you can use to do direct comparisons with compounds 5. This would help them to discuss some SAR. Really, if the authors wished to discuss SAR they should choose more carefully the substrates that are synthesized and also make additional analogues where they do strategic manipulations at certain places of the compounds so a direct comparison to be possible.

With these limited data I find that the conclusions drawn in lines 159-167 have no meaning. Statements 1-5 should be removed or rephrased.

Line 159 “A thiazole ring is essential for cytotoxic activity”: how is this concluded since all the compounds contain a thiazole ring? No other rings are studied so we don’t know if in the place of the thiazole, there was an imidazole or an isoxazole would exhibit the same or better activity. As such this statement has no real meaning and should be removed.

Line 161: Rephrase to something like “the analogue 5b was more cytotoxic than compound 5a”

Line 163: “Position 5 of the thiazole ring is the best place for substitution” I don’t know what is meant by that. Please remove.

Line 164: “The presence of an acetyl group at position 5 of the thiazole ring enhances potency and toxicity.” Which is the reference compound for this comparison? I don’t understand which compounds are compared here. Are they comparing the C=O vs C=N? I think that this comment has also no meaning and should be removed.

Line 166: Same as above, regarding to what are the authors comparing to?

In general, I don’t find any rationality in statements 1-5 and owing to the limited data no conclusions of structure activity relationships can be drawn. As such Figure 3 should also be removed.

Experimental section:

In the general procedure, in lines 217-220, some information is missing and is needed to be included: What is the concentration of the ethanolic solution (i.e. the volume of EtOH used). For all the reagents that are constant (i.e. all the reagents except the aniline derivative) the mass or volume should be included in parenthesis together with the mmols.

In the spectroscopic characterisation of each individual compound (for all the synthesized compounds 2, 5, 7 and 8):

(a)   In addition to the yield % the mass in mg obtained should be included for each compound.

(b)   the recrystallisation solvent should be mentioned in parenthesis after the mp

(c)   the mp should be given as a range and not a single value.

(d)   A description of the physical form of compound should be made e.g. obtained as colorless needles, or yellow prisms, or colorless plates etc (this description should be included before the yield).

(e)   For compounds 2c the 13C is missing

(f)    All the compounds 2a-e are known in the literature. The relevant literature should be cited and comparison with the literature spectroscopic data should be made. Ideally the literature mp should be included in parenthesis after the authors mp for comparison and a line at the end stating that the “spectroscopic data are in agreement with the literature” (if they are)

Refs: compd 2a-b: Phosphorus, Sulfur, and Silicon, 2011, 186, 12–20.

2c-d: J. Am. Chem. Soc., 1953, 75, 102–104.

2e: Reaxys gives this refs: Arkiv foer Kemi1956, 9, 127; Arkiv foer Kemi1955, 7, 249 but I don’t have access.

(e) compd 7 it is known in the literature and the comparison to the literature should be made Ref: Russian Journal of Organic Chemistry2007, 43, 1516. Interestingly the mp reported in the literature it is much different from the one reported here (lit. 155-157 °C vs 285 °C). I highly recommend that the authors very carefully check all their spectroscopic data vs the known data for this compound.

(f) the 13C-NMR for compound 7 is missing.

(g) compd 8 is known. The same things as mentioned above for the known lit compounds applies here too. The relevant literature, Journal of Chemical Research, Miniprint 1994, 4, 747-760, should be cited and comparison of the data should be made.

Since the compounds are used for biological studies their NMR spectra should be provided in the SI for further support of their purity.

The authors provide the elemental analysis which is very good.

Formatting/grammar/typographical errors

The manuscript contains a huge number of typographical/formatting/grammar errors that need to be corrected. In many cases the compounds numbers are not in bold. Please proofread all the manuscript and change all the compound numbers in bold.

some corrections include the following:

Line 33 and 39: change ‘synthesizing’ to ‘synthesis’

Line 51: change ‘resulted’ to ‘results’ and ‘reacted’ to ‘react’

Line 53: change ‘yielded’ to ‘yields’

Line 57: change to ‘spectral’ to ‘spectroscopic’

Line 62: change ‘The refluxing of compound 2a’ to ‘Refluxing compound 2a’

Line 65: change ‘became’ to ‘was’

Line 67: chane ‘]’ to ‘)’

Line 68 and 86-90: ‘Error! Reference source not found.’ Please correct this

Line 72, 215, 278: change ‘[6]’ to ‘(6)’

Line 124 and 304: correct ‘zoon’ to ‘zone’

Line 177: change ‘good biological attitude’ to ‘promising biological profile’

References: Please proofread very carefully all your references, there are many mistakes. Also, in some cases issues are included in parenthesis, these should be removed. In other cases, the full journal title is used instead of the abbreviation. Please correct accordingly.

Some of the mistakes are mentioned below:

Line 333:B.S.4-Thiazolidones:” missing space

Line 343: ‘J. Chem. Soc.: Perkin Trans’ correct to ‘J. Chem. Soc., Perkin Trans 2’

Lines 352-353: ‘Journal of Pharmacology and Experimental Therapeutics’ correct to the cassi abbreviation ‘J. Pharmacol. Exp. Ther.’

Line 257: ‘J Med Chem.’ missing full stops.

Line 364: ‘studies’ to ‘Studies’

Line 374:effectofhKv1.5channel. Arch Pharm Res 2007, 30 (2), 155-160.’ Missing spaces and full stops, correct toeffect of hKv1.5 channel. Arch. Pharm. Res. 2007, 30, 155-160.

Line 376: correct journal abbreviation name toJ. Pestic. Sci.’

Line 379: correct journal title to ‘Acta Pol. Pharm.

Line 382. Missing full stops in journal abbreviation

Line 409: not the full title is given. correct the article title to ‘A one-pot synthesis of N-alkylthiazoline-2-thiones from CS2, primary amines, and 2-chloro-1,3-dicarbonyl compounds in water’

Line 412: remove ”. correct journal name to ‘Green Sustainable Chem.’

Line 414-415: correct journal name to the abbreviated form

Line 421-422: correct title to ‘Tables of bond lengths determined by X-ray and neutron diffraction. Part 1. Bond lengths in organic compounds’ and correct the rest to ‘J. Chem. Soc., Perkins Trans. 2 1987, 1-19.’

Line 428: missing full stops in abbreviation

Line 432: remove ‘(2008)’

Line 432: correct journal name to abbreviation ‘Acta Crystallogr., Sect. A: Found. Adv.’

To conclude the manuscript suffers from poor presentation, omitting to mention that the majority of the compounds are known in the literature, not all the relevant literature is cited and there are not sufficient data to support the SAR comments that the authors make regarding the biological activities (antitumor studies for at least compd 2a are required.  And why not for the rest of the compounds 5 -  or the authors should explain the rational for performing the studies for only selected examples of the analogues 5). 

These weaknesses need to be addressed before publication.

Author Response

(The authors gave the same response as above.)

Reviewer 3 Report

 The chemical section is well described, but the table concernong the biological resluts must be improved, in this form is very difficult to read it.

Anyway, attached you can find the PDF with several suggestions

Author Response

Dear Editor:

Thank you for your E-mail concerning manuscript number (molecules-479281).  Attached herewith, please find our revised manuscript having complied with technical remarks of the reviewers. We have complied with all requested corrections. Key changes are noted in red in the revised Manuscript. Further, the authors provide detailed responses to referee comments, on the pages that follow this cover letter. These comments are also recorded in red. On behalf of the coauthors, I would like to thank the reviewers for their feedback and constructive comments

I sincerely hope that the amended version is satisfactory for publication.

Thanking you for your cooperation and best regards.

Sincerely, yours

Yahia N. Mabkhot

Professor of Organic Chemistry

Department of pharmaceutical Chemistry

College of pharmacy

King Khalid University, Abha 61441, Saudi Arabia;

Round 2

Reviewer 1 Report

In the revised form of the paper "A Convenient Synthesis, X-Ray Analysis, Biological Evaluation and Molecular Docking Study of New Thiazole Derivatives" by Yahia N. Mabkhot et al., the authors did some modifications, but they skipped the most important ones:

1. the Introduction: even with all the new references added, it still remains unclear; in addition, it is not really linked to the title, from where a reader would expect the paper to be about thiazolines (about what it actually is), while the introduction is exclusively about thiazoline-thiones, and mostly about their synthesis, nothing about the antimicrobial activity of these compounds, nor about the antiproliferative effect or molecular docking studies; hence, the short dimension of this part;

2. concerning the antibacterial and antitumor activities investigation: "we evaluated the antimicrobial/antitumor activity for some selected examples to assess their activity. Complete study for their antimicrobial activity will be included in another study" is a very irreponsible reply; you can not publish the synthesis of 20 compounds, but the biological evaluation of only 10, without mentioning the criteria of selection (justifying) or the reason why the other 10 were excluded; and "these results will be published in another study" is not a good reason;

3. same opinion about the answer concerning my request about the reference drugs (doxorubicine, ampicillin, gentamycin and amphotericin): "others used them too" is not an answer; I am sure that the "other authors" justified their choice;

4. about the molecular docking study: I believe the authors are beginners in this field (no offense); if you use a certain anticancer drug for the in vitro tests, you have to use it's target for the molecular docking study, otherwise the whole comparison or structure-activity discussion makes no sense; and the reference drug should always be included in the docking study, along with the compounds; and again " we will use doxorubicine in another target" is not the right answer. 

Author Response

Dear Editor:

Thank you for your E-mail concerning manuscript number (molecules-479281).  Attached herewith, please find our revised manuscript having complied with technical remarks of the reviewers. We have complied with all requested corrections. Key changes are noted in red in the revised manuscript. On behalf of the coauthors, I would like to thank the reviewers for their feedback and constructive comments

I sincerely hope that the amended version is satisfactory for publication.

Thanking you for your cooperation and best regards.

Sincerely, yours

Yahia N. Mabkhot

Professor of Organic Chemistry

Department of pharmaceutical Chemistry

College of pharmacy

King Khalid University, Abha 61441, Saudi Arabia;

Reviewer 2 Report

Many thanks to the authors for their noteworthy effort to accommodate all the reviewers' comments. 

I am happy with the changes. 

I have some additional minor corrections: 

1) In Scheme 1 in the table the yields for the products 2a-e  need to be included 

2) in line 225 correct 'Etanol' to 'EtOH'

3) In line 272 after the lit mp include the lit recryst solvent in parenthesis i.e. correct to

'[Lit mp. 155-157 oC (MeCN)][40]'

4) line 315: correct the abbreviation to 'Spectrochim. Acta, Part A: Mol. Spectrosc.'

5) In general check all your references. many abbreviations are written without the full stops, e.g. you write 'J Org Chem' instead of  'J. Org. Chem.

please correct all these instances accordingly. and check the refs abbreviations to be in agreement  with http://cassi.cas.org/search.jsp

6) line 406: the volume and page range given are those for the Russian version. 

please correct it to the English version which is: vol 43 and pages: 1516-1525

i.e. correct: Russ. J. Org. Chem. 2007, 43, 1516-1525.

Author Response

(The authors gave the same response as above.)
